# A Postearthquake Multiple Scene Recognition Model Based on Classical SSD Method and Transfer Learning

**Zhiqiang Xu, Yumin Chen \*, Fan Yang, Tianyou Chu and Hongyan Zhou**

School of Resource and Environment Science, Wuhan University, Wuhan 430079, China;
xuzq97@whu.edu.cn (Z.X.); fan_yang@whu.edu.cn (F.Y.); chutianyou@whu.edu.cn (T.C.);
2019282050143@whu.edu.cn (H.Z.)

\* Correspondence: ymchen@whu.edu.cn

**Abstract:** The recognition of postearthquake scenes plays an important role in postearthquake rescue and reconstruction. To overcome the over-reliance on expert visual interpretation and the poor recognition performance of traditional machine learning in postearthquake scene recognition, this paper proposes a postearthquake multiple scene recognition (PEMSR) model based on the classical deep learning Single Shot MultiBox Detector (SSD) method. In this paper, a labeled postearthquake scenes dataset is constructed by segmenting acquired remote sensing images, which are classified into six categories: landslide, houses, ruins, trees, clogged and ponding. Due to the insufficiency and imbalance of the original dataset, transfer learning and a data augmentation and balancing strategy are utilized in the PEMSR model. To evaluate the PEMSR model, the evaluation metrics of precision, recall and F1 score are used in the experiment. Multiple experimental test results demonstrate that the PEMSR model shows a stronger performance in postearthquake scene recognition. The PEMSR model improves the detection accuracy of each scene compared with SSD by transfer learning and data augmentation strategy. In addition, the average detection time of the PEMSR model only needs 0.4565s, which is far less than the 8.3472s of the traditional Histogram of Oriented Gradient + Support Vector Machine (HOG+SVM) method.

**Keywords:** earthquake disasters; scene recognition; deep learning; classical SSD method; transfer learning

## 1. Introduction

Earthquakes are one of the most harmful types of natural disasters in the world. Approximately five million earthquakes occur every year worldwide, of which about a dozen or twenty have caused serious harm to humanity, resulting in incalculable environmental damage and loss of life and wealth. Take the 2014 magnitude 6.5 Ludian earthquake as an example: it caused a death toll of 617, triggered at least 1024 landslides with areas equal to 100 m$^2$ or larger and tens of thousands of collapsed buildings [1,2]. The quick and accurate collection of damage information in earthquake-stricken areas is of substantial significance for the timely rescue of trapped people and postearthquake reconstruction [3,4]. In seismic emergency rescue work, the most traditional method is onsite investigation by relevant experts [5,6]; however, the workload is extremely large, and the efficiency is low due to the large extent and variety of disaster areas [7]. It is difficult to reach the disaster sites in time if investigators encounter landslides or clogged scenes. Due to low efficiency and uncertainty, it is not currently possible to satisfy the application requirements of rapid assessment and postearthquake rescue.

However, with the rapid development of technologies such as satellite remote sensing and unmanned aerial vehicles, the ability to acquire real-time information on the Earth's surface has

improved [8,9]. Remote sensing images can be acquired quickly and can reflect the objective world comprehensively and intuitively, and they provide a new information source for the rapid recognition and assessment of earthquake damage [10,11]. There is a lot of research about disaster risk assessment based on remote sensing images. Jelének et al. [12] synergically used Sentinel-1 radar images and Sentinel-2 optical data to analyze postearthquake surface changes and took the 2016 magnitude 7.8 Kaikoura earthquake in New Zealand as an example. They used radar interferometry to assess earthquake impacts via computing vertical displacements and differential interferograms. Olen et al. [13] proposed a new method for Potentially Affected Area (PAA) detection following a natural hazard event based on Sentinel-1 C-band radar data. The proposed method is based on the coherence time series, which determines the natural variability of coherence within each pixel in the region of interest and where statistically significant coherence loss has occurred by comparing pixel-by-pixel syn-event coherence to temporal coherence distributions. They verified the performance of the method in finding PPA with the case of the 2017 Iran–Iraq earthquake and a landslide-prone region of NW Argentina. Mondini et al. [14] proposed that using Sentinel-1 SAR C-band images could well solve the problem of lack of pre and postlandslide optical images due to cloud persistence. They analyzed 32 global landslide cases, and results showed that changes caused by landslides on SAR amplitudes were unambiguous in about 84% of cases. Expert visual interpretation methods that fully utilize high-resolution images have become mainstream in the field of postearthquake assessment, rescue and reconstruction [15–17]. However, these methods suffer from inefficiency and high costs in terms of expert resources. Moreover, the interpretation of results differs substantially across experts [18,19].

In recent years, the development of machine learning has helped to overcome some of these limitations by promoting the use of computer image recognition and processing [20,21]. Furthermore, with the development of technologies such as GPU and artificial intelligence, image recognition via deep learning methods has become more efficient and accurate [22,23], which enables the use of deep learning to realize postearthquake scene recognition. The core steps of image recognition are typically feature extraction and classification. In the early days, image recognition mainly used traditional manual feature extraction methods, such as Scale-invariant Feature Transform (SIFT) [24], Histogram of Gradient (HOG) [25] and Deformable Parts Model (DPM) [26], in combination with classifiers such as Support Vector Machine (SVM) [27] and random forest [28]. Since Hinton [29] proposed a solution to the problem of gradient disappearance in deep network training, deep learning entered a period of substantial development. After Convolutional Neural Networks (CNNs) [30] were proposed in 2012, deep learning was developed explosively; CNN has been fully developed and has been applied to many research fields. There are two typical types of deep learning for image recognition: methods that are based on region proposal, such as RCNN [31], FAST-RCNN [32], FASTER-RCNN [33] and R-FCN [34] and methods that are based on regression, such as You Only Look Once (YOLO) [35] and Single Shot MultiBox Detector (SSD) [36]. The methods of the second type are faster but less accurate than those of the first type, because they generate bounding boxes in a single net. Compared to YOLO, the SSD method not only improves the speed but also improves the recognition accuracy, which is comparable to the RCNN series [36]. Therefore, the SSD method is adopted in our model.

Recently, many researchers have applied deep learning methods to disaster scene recognition. Ding et al. [37] considered a Google postearthquake image with a spatial resolution of 0.3 m in Ludian county, Yunnan province of China as an example and used a pretrained AlexNet deep convolution neural network model for feature extraction, in combination with a SVM classifier, to realize postearthquake scene recognition. Sun et al. [38] proposed a convolutional neural network that was combined with multiscale segmentation (CMSCNN) for high-resolution seismic image classification, which realized improved accuracy. Xu et al. [39] developed a Dense Feature Pyramid model with an encoder–decoder network (DFPENet) for coseismic landslide recognition, and the experimental results demonstrated its high-precision, high-efficiency and cross-scene recognition of earthquake disasters. Ji et al. [40] proposed a CNN feature with the random forest method; compared with CNN, this method improves

the accuracy of postearthquake collapse identification and the feature extraction performance of CNN is better than that of texture feature extraction. Song et al. [41] proposed a method that used the Deeplab v2 neural network for the initial identification of damaged building areas and applied the simple linear iterative cluster (SLIC) method to accurately extract the area boundaries of the earthquake-damaged buildings. Finally, a mathematical morphological method was introduced for eliminating the background noise in this paper.

The methods that are discussed above yielded substantial results in the field of postearthquake scene recognition through the optimization of network structure and integration with other algorithms. However, these methods rarely consider the lack of data and may struggle to perform well with insufficient data. Especially in postearthquake remote sensing image recognition, a substantial obstacle is the lack of labeled samples. Therefore, it is important to establish a postearthquake scene recognition model that can perform well with only a small amount of data.

In this paper, a postearthquake multiple scene recognition (PEMSR) model based on the classical SSD detection method and transfer learning is proposed. The model attempts to collect postearthquake scenes images and to label them manually for the construction of a dataset. To eliminate the negative influence of an insufficient dataset, data augmentation and transfer learning [42] are used in this model. In addition, random oversampling is utilized to overcome the problem of data imbalance. The PEMSR model and other models are evaluated and compared to examine the model's performance and the impacts of data augmentation and transfer learning on the PEMSR model.

## 2. Materials and Methods

The PEMSR model's workflow is illustrated in Figure 1, which has five major components: (1) data collection; (2) data preprocessing; (3) transfer learning with SSD; (4) testing and validation; and (5) model evaluation.

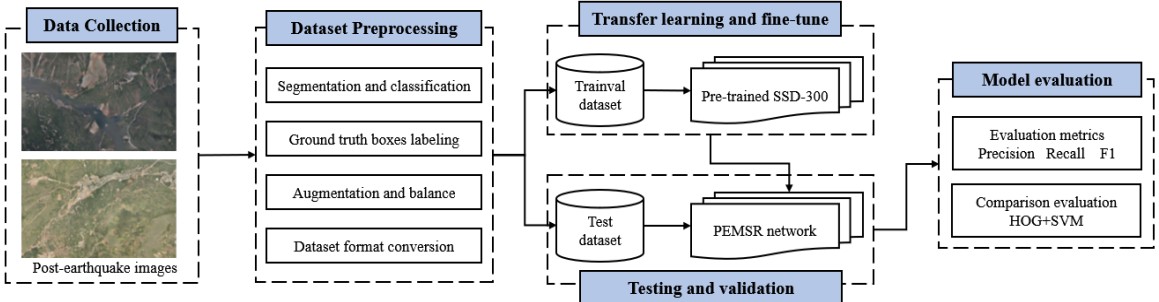

**Figure 1.** PEMSR model's workflow. Trainval dataset means training and validation dataset.

### 2.1. Data Collection

In this paper, Google images of the earthquake-stricken area of Ludian County, Zhaotong City, Yunnan Province of China on August 7, 2014 with a spatial resolution of 0.3 m are collected as the raw data for recognition. Detailed information is presented in Table 1.

**Table 1.** Image information.

| Date | Location Name | Spatial Resolution | Image Size |
|---|---|---|---|
| 2014/08/07 | Niulan River | 0.3 m | 44,667*30,000 |
| 2014/08/07 | Longtou Mountain | 0.3 m | 89,387*80,000 |

### 2.2. Data Preprocessing

After the images have been obtained, it is necessary to preprocess the raw data so that it can be adapted for network training. The main steps of data preprocessing are (1) data segmentation and

classification; (2) ground truth boxes labeling; (3) data augmentation and balance; and (4) dataset format conversion.

### 2.2.1. Data Segmentation and Classification

In this phase, the images which had wide ranges were too large to train and test. Hence, we segmented images through ArcGIS Desktop to extract useful scenes for the experiment. Each image is cropped to close to 300*300 pixels, which is convenient for the subsequent processing. The segmented images are classified manually as six types of scenes: landslide, ruins and clogged which are caused by an earthquake, along with common scenes, namely, houses, ponding and trees. These six types of scenes constitute our original dataset. The number of instances of each type of scene is specified in Table 2. Considering the limited amount of data in the original dataset, in order to ensure sufficient training dataset to avoid overfitting and the confidence of the test results, the segmented images are randomly divided into a trainval (training and validation) dataset and a test dataset in a 4:1 ratio. As the original images are too big to present, parts of the original images and the segmented postearthquake scene images are shown in Figure 2.

**Table 2.** Number of instances of each scene.

| Scene | Houses | Landslide | Ruins | Ponding | Trees | Clogged |
|-------|--------|-----------|-------|---------|-------|---------|
| Num. | 125 | 52 | 115 | 21 | 26 | 24 |

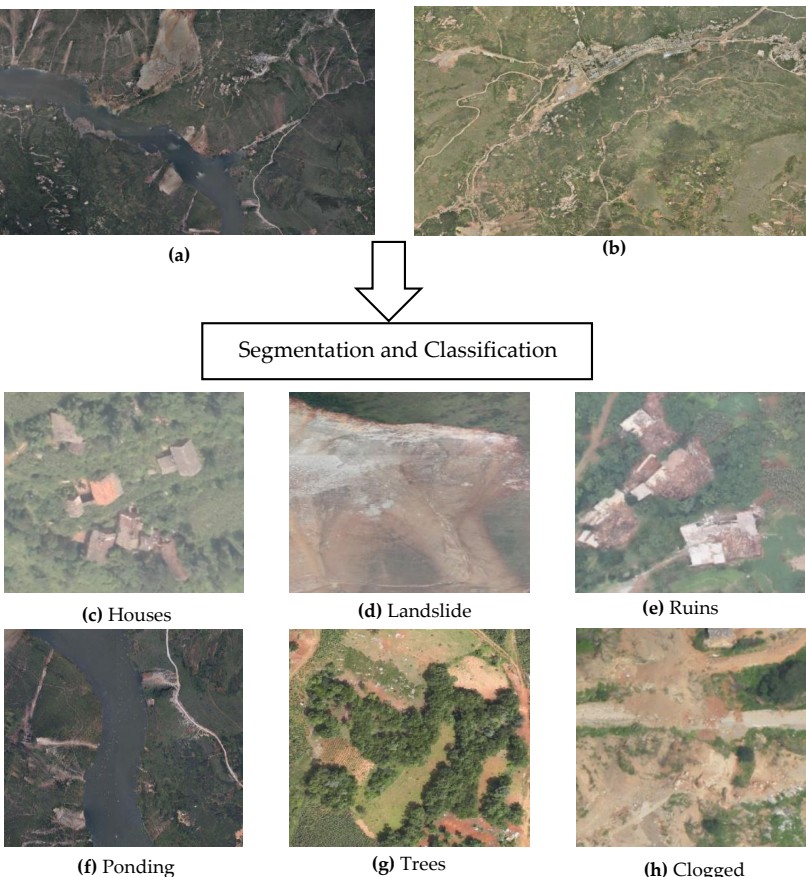

**Figure 2.** Results of segmentation and classification. Image (**a**) is of part of Niulan river and image (**b**) is of part of Longtou mountain. Segmented and classified postearthquake scenes: (**c**) houses, (**d**) landslide, (**e**) ruins, (**f**) ponding, (**g**) trees and (**h**) clogged.

2.2.2. Ground Truth Boxes Labeling

Classified images with manually labeled of regions of interest (ROIs) are necessary as prior knowledge in the experiment. The ROIs of every image are delineated by the blue ground truth boxes, and TXT files are generated for recording the boxes' information, where each TXT file corresponds to an image. The TXT file's format is as follows:

Object Number
ClassName x1min y1min x1max y1max
ClassName x2min y2min x2max y2max

The number of ground truth boxes and each box's name and location in the corresponding image are recorded. A labeling result example is presented in Figure 3.

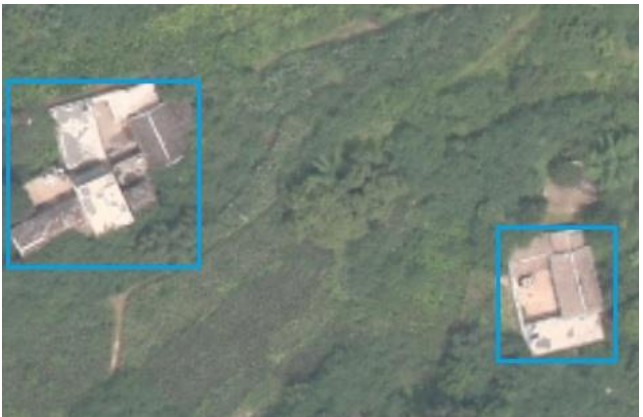

**Figure 3.** Labeling result of an image. Blue boxes indicate the real location of houses, and their information is recorded in a TXT file.

2.2.3. Data Augmentation and Balance

In the deep learning method, fewer training samples may result in insufficient learning performance of the model, poor generalization of the model and overfitting. To overcome the problem of insufficient sample data in the experiment, we apply the data augmentation [43,44] method of rotation and mirroring to obtain additional similar sample data. As presented in Table 3, the mirroring transformation doubles the number of original sample data. The rotation transformation is the rotation of the original images by 90, 180 and 270°, which can increase the number of original sample data four times. In the experiment, this paper combines the two data augmentation methods and uses them to expand the sample size to 2, 4 and 8 times the original sample size. In the data augmentation process, the labeled ground truth boxes are augmented simultaneously, and new corresponding annotation TXT files will also be generated.

**Table 3.** Data augmentation method.

| Augmentation Method | Transformation Parameter |
| --- | --- |
| Rotation | 90°, 180°, 270° |
| Mirroring | (x, y) = (y, x) |

In addition, there is a severe imbalance in the distribution of the original scenes, which may impact the performance of the model. There are three main approaches to deal with class imbalance problems, which can be classified as data-level, algorithm-level and hybrid methods. To ameliorate the sample imbalance problem in the dataset, the random oversampling method [38] is used to replicate and increase the number of samples in minority classes in this paper. Table 4 presents the data volumes

after data augmentation and data balance; the number of scenes in the dataset increases and tends to be more balanced. After each data augmentation, we ensured that the ratio of the trainval (training and validation) dataset to the test dataset is 4:1. After the last data augmentation and balance, the total number of trainval dataset is 4608 and the total number of test dataset is 1152.

**Table 4.** Data volume after augmentation and balance. Two times the original number of data is obtained via the mirroring transform, four times via the rotation transform and eight times via the combination of the two transforms. The last row is the data distribution after data balance.

| Times | Houses | Landslide | Ruins | Ponding | Trees | Clogged |
|---|---|---|---|---|---|---|
| Original | 125 | 52 | 115 | 21 | 26 | 24 |
| Twice | 250 | 104 | 230 | 42 | 52 | 48 |
| Four | 500 | 208 | 460 | 84 | 104 | 96 |
| Eight | 1000 | 416 | 920 | 168 | 208 | 192 |
| Balanced | 1000 | 832 | 920 | 1008 | 1040 | 960 |

### 2.2.4. Dataset Format Conversion

In this paper, all experiments are based on a deep learning open source framework, namely, Caffe [45]. Caffe requires the training data format to be LMDB; hence, it is necessary to convert the dataset to LMDB format before training. The main steps are as follows:

Create four new folders: Annotations, JPEGImages, ImageSets and Labels. All images are numbered and stored in the JPEGImages folder, and corresponding TXT label files are stored in the Labels folder.

Convert the TXT label files to XML format using a script and store them in the Annotations folder.

Run a script to generate trainval (training and validation) dataset and test dataset identification files and store them in the main folder of ImageSets. Via this approach, the dataset is converted into VOC format.

Modify and run the conversion script to convert the VOC dataset to LMDB format. The formatted trainval (training and validation) and test datasets are generated and stored in the LMDB folder. Data preprocessing is completed.

### 2.3. Transfer Learning with SSD

Transfer learning has the advantages of low data requirements, flexibility and robustness, which can improve the PEMSR model training efficiency and accuracy. To overcome the lack of sufficient labeled samples, this paper utilizes transfer learning [42,43,46] with a pretrained model that is obtained via SSD training on the PASCAL VOC dataset. There are two types of SSD structures that are most widely used, including SSD-300 and SSD-512. This paper chooses the SSD-300 as the base model, which also fits the images size of our dataset 300 * 300. The SSD-300 [36,47] method consists of two components: a deep convolutional neural network based on the VGGNet-16 network structure [48] for target preliminary feature extraction and a multiscale convolutional feature detection network. This network combines multiscale feature maps and convolution operators to generate bounding boxes with probabilities that they contain the objects of interest. Then, the final recognition results are obtained via nonmaximum suppression (NMS). Our PEMSR model inherits the advantages of this structure and fine tunes network parameters by learning from our postearthquake scenes dataset.

Fine-tuning is an important skill for transfer learning. As shown in Figure 4, the parameters of the PEMSR model are initialized by the parameters of the SSD-300 model that is pretrained on the PASCAL VOC dataset. At the same time, according to the input postearthquake scenes dataset, the weights of the layers are fine-tuned for class prediction and the generation of bounding boxes. In the model training phase, the batch size is set to 64, the weight decay is set to 0.05, the momentum is set to 0.9, the number of iterations is set to 24,000 and the base learning rate is set to 0.01 and divided by 10 when the number of iterations reaches 8000 and 16,000.

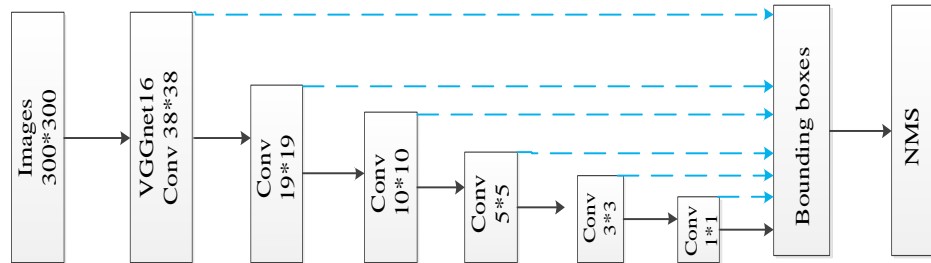

**Figure 4.** PEMSR network proposed in this paper. The black arrows represent the fixed initial parameters that are learned on the PASCAL VOC dataset, and the blue arrows represent the weights that are learned on this paper's dataset. The sizes of the feature maps in the five subsequent convolutional layers are 19, 10, 5, 3 and 1.

*2.4. Testing and Validation*

The dataset of this paper is randomly divided into ten subsets, with seven subsets used as the training dataset, one subset used as the validation dataset and the remaining subsets used as the test dataset. After the PEMSR training model was obtained, the test script was run and the test result was obtained. Figure 5 presents a test recognition result; in the upper left corner of the box, the class name and the confidence value are specified.

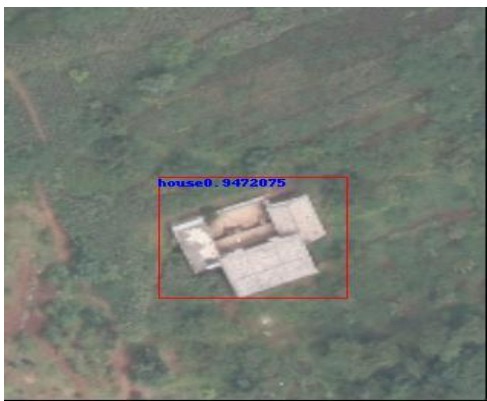

**Figure 5.** Example of a test result.

*2.5. Model Evaluation*

2.5.1. Evaluation Metrics

In this paper, the precision, recall and F1 are selected as evaluation metrics. The precision is the ratio of the number of positive samples that are identified correctly as positive samples. The recall reflects whether all positive examples are recognized. The F1 score is the harmonic mean of the recall and the precision, which is a more comprehensive evaluation metric; a value that exceeds 0.6 is considered satisfactory. These metrics are calculated as follows:

$$precision = \frac{tp}{tp + fp} \tag{1}$$

$$recall = \frac{tp}{tp + fn} \tag{2}$$

$$F1 = \frac{2 * P * R}{P + R} \tag{3}$$

where $tp$ is the number of positive samples that are correctly identified as positive, $fp$ is the number of negative samples that are incorrectly identified as positive, $fn$ is the number of positive samples that are not recognized, $P$ is the precision value and $R$ is the recall value.

2.5.2. Comparison Evaluation with the HOG+SVM Method

For further comparison and evaluation of the PEMSR model, this paper considers the traditional target recognition method HOG+SVM in this experiment. HOG is used for feature extraction and SVM is used for feature classification.

The HOG+SVM [25,27,49–51] method mainly includes the following steps:

1)  Image graying: convert RGB images to grayscale;
2)  Color normalization: normalize the color space by the gamma correction method;
3)  Gradient calculation: calculate the gradient of each pixel of the image;
4)  Cells segmentation: divide the image into small 6 * 6 cells and statistic gradient histogram;
5)  Block descriptor: combine 2 * 2 cells into a block to get the block descriptor;
6)  Block descriptor normalization: normalize contrast for each block and collect HOGs over detection window;
7)  SVM classifier: send the HOG feature vector to the SVM classifier.

Besides, the standard SVM is a binary classifier for pattern recognition, and it must be extended to a multiclass classifier for learning a multiclass problem. Hence, this paper combines the two types of postearthquake scenes to form 15 new classifiers. For convenience, the six types of scenes are denoted by A, B, C, D, E and F. Each training on unknown samples will yield 15 training results. In the test, the corresponding vector of unknown samples is tested on the 15 results; then, voting is conducted for classification. The complete voting process is as follows:

First, A = B = C = D = E = F = 0;
In the (A, B) classifier, if A wins, then A = A + 1; otherwise, B = B + 1;
In the (A, C) classifier, if A wins, then A = A + 1; otherwise, C = C + 1;
In the (D, F) classifier, if D wins, then D = D + 1; otherwise, F = F + 1;
In the (E, F) classifier, if E wins, then E = E + 1; otherwise, F = F + 1;
Finally, after 15 comparisons, the classification result of the image is Max (A, B, C, D, E, F).

## 3. Experiments and Results

This section introduces the study area, experimental environment preparation and design methods that are utilized in this paper. Multiple sets of comparison and optimization experiments were conducted to identify the optimal PEMSR model.

### 3.1. Study Area

This paper selected the 2014 magnitude 6.5 Ludian earthquake area as the study area, which is related to the earthquake time and location. There are several important characteristics of the study area. Firstly, Ludian County was a poverty-stricken county at the national level, located in the northeast of Yunnan, China. The local economic conditions were relatively poor and the seismic resistance of buildings were generally poor, so that too many buildings collapsed when the earthquake happened. Secondly, the earthquake area was mountainous and coincided with the rainy season so that caused more serious secondary disasters, such as landslides and mudslides. These scenes are more obvious and typical on remote sensing images with complex backgrounds, so this paper selected Ludian earthquake area as the study area.

### 3.2. Experiment Preparation and Design

Our experiments are conducted on an Ubuntu 14.04 LTS (64-bit) operating system. We train the model on a GTX 1080 with 8G of memory; other environmental configurations are presented in Table 5.

For evaluating the performance of the PEMSR model and the impacts of transfer learning, data augmentation and balance on the model more extensively, this paper designed three levels

of experiments: first, comparison with the HOG+SVM and initial SSD methods to evaluate the performance of the three models on postearthquake scene recognition; second, after data augmentation by 2, 4 and 8 times, comparison of the performance of the PEMSR model before and after data augmentation; and third, comparison of the performance of the model before and after data balancing.

**Table 5.** Experimental environment configuration.

| Environment | Version |
|---|---|
| Operating System | Ubuntu 14.04 LTS (64-bit) |
| Deep Learning Framework | Caffe |
| Memory | 64G |
| GPU | Nvidia GTX 1080 |
| CPU | Intel(R) Core (TM) i7-6800k |
| Library | CUDA 8.0, CuDNN 8.0 |

*3.3. Experimental Results*

3.3.1. Comparison of the Results of Three Methods

The HOG+SVM method and the SSD [36] method of initial parameters are compared with the PEMSR model for a comparison evaluation. The F1 score results of these methods are presented in Table 6. According to Table 6, the F1 scores of the HOG+SVM method are only slightly higher than those of the SSD and PEMSR methods for the ponding and trees scenes, while the F1 scores for the other scenes are far lower than those of the other two methods. The F1 scores of the SSD and PEMSR models on houses, landslide, ruins and ponding exceed 60%; the scores do not reach 60% on trees and clogged which points to the need for further improvement. In addition, the average detection times of both the SSD and PEMSR methods are shorter than 0.5s; these methods are much faster than the traditional HOG+SVM recognition method. In summary, deep learning methods outperform the traditional machine learning method overall. Comparing the performance of SSD and PEMSR, although the SSD method is slightly faster in terms of the detection time, PEMSR realizes higher recognition accuracy on several scenes, which reflects the positive influence of transfer learning.

**Table 6.** F1 score results of the three methods on the original data. HS refers to the F1 score results of the HOG+SVM method, ADT refers to the average detection time and the bold values are the best results among the three methods.

| Method | Houses | Landslide | Ruins | Ponding | Trees | Clogged | ADT |
|---|---|---|---|---|---|---|---|
| HS | 64.86% | 33.33% | 51.85% | **80.00%** | **40.00%** | 26.92% | 8.3472 s |
| SSD | 75.22% | 83.57% | 82.95% | 70.31% | 37.88% | 46.21% | **0.4123 s** |
| PEMSR | **79.17%** | **86.67%** | **87.78%** | 72.73% | 39.98% | **50.00%** | 0.4565 s |

3.3.2. Results Comparison after Data Augmentation

To optimize the PEMSR model, a data augmentation strategy is used in this paper. The F1 score results that are obtained after augmentation of the data 2, 4 and 8 times are presented in Table 7. The overall recognition accuracy of the PEMSR model increases as the samples' volume increases, especially for the trees and clogged scenes, which occupy only small parts in the original dataset. Through data augmentation, the recognition accuracy of each scene has been improved and all scenes' F1 scores exceed 60%. However, this does not imply that as the sample size increases, the recognition performance will improve indefinitely. For example, for the ruins data with augmentations of four and eight times in Table 7, the amount of data increased, but the F1 score declined slightly. It is believed that the optimal recognition performance has been approached or realized on the ruins samples; the best recognition performance is realized by the PEMSR model, and it is difficult to improve via data augmentation.

**Table 7.** F1 score results of the PEMSR method under various data volumes. The bold values are the best results for each scene.

| Times | Houses | Landslide | Ruins | Ponding | Trees | Clogged |
|---|---|---|---|---|---|---|
| Original | 79.17% | 86.67% | 87.78% | 72.73% | 39.98% | 50.00% |
| Twice | 80.00% | 86.24% | 87.34% | 72.79% | 53.33% | **63.16%** |
| Four | 80.61% | 85.48% | **88.06%** | 74.42% | 57.14% | 60.00% |
| Eight | **88.28%** | **90.37%** | 88.02% | **76.08%** | **64.10%** | 62.06% |

### 3.3.3. Results Comparison after Data Balancing

The imbalance of the dataset may make the performance of the PEMSR model biased toward minority samples. Several experiments were conducted to evaluate this hypothesis. After we used random oversampling to make data samples more balanced, the recognition F1 score results presented in Table 8 were obtained. They are performed on the original, augmented and balanced samples. The F1 score results of data augmentation are the optimal recognition accuracy on each scene in the previous augmentation experiments. After the samples are balanced, the F1 scores on the ponding, trees and clogged scenes, which occupy relatively small parts of the previous dataset, increase substantially; those for the other scenes do not change substantially.

**Table 8.** F1 score results of the PEMSR model under various conditions. SI refers to the best F1 score results after the sample volume was increased, and SB refers to the F1 score results after the sample was balanced. The bold values are the best results among several experiments for each scene.

| Data | Houses | Landslide | Ruins | Ponding | Trees | Clogged |
|---|---|---|---|---|---|---|
| Original | 79.17% | 86.67% | 87.78% | 72.73% | 39.98% | 50.00% |
| SI | **88.28%** | 90.37% | 88.06% | 76.08% | 64.10% | 63.16% |
| SB | 87.88% | **92.32%** | **89.02%** | **83.66%** | **77.34%** | **72.16%** |

### 3.3.4. Optimal PEMSR Model

Based on the results of comparative experiments, the optimal PEMSR model is regarded as the model with data augmentation by a factor of eight and data balancing. According to Figure 6, it can be seen that the loss value decreases rapidly in the first 50 epochs and then gradually flattens until it converges. Test results of the optimal PEMSR model are presented in Figure 7, and its precision, recall and F1 score are presented in Table 9. All these values exceed 70% on all scenes and exceed 90% on the landslide scene. Hence, satisfactory recognition performance has been realized.

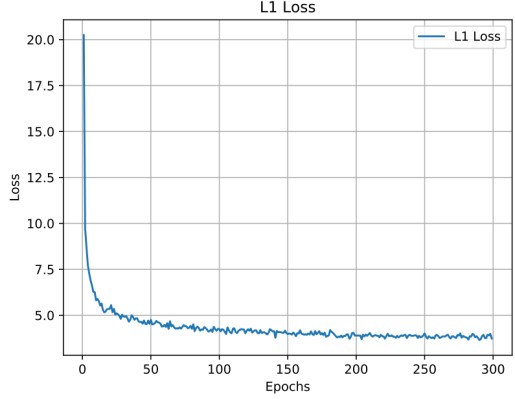

**Figure 6.** Relation between epochs and loss value of optimal PEMSR model.

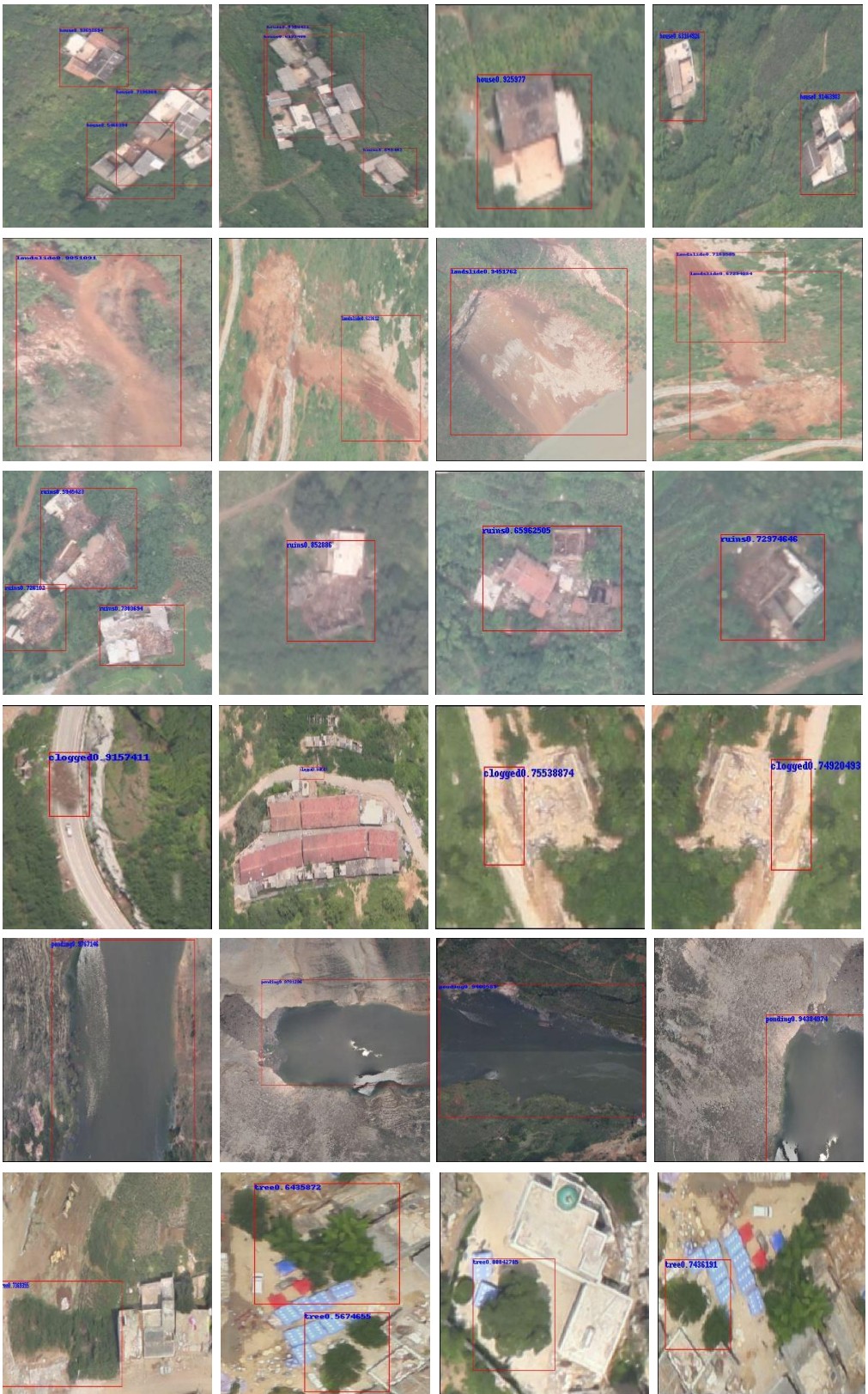

**Figure 7.** Examples of optimal PEMSR model test results. From top to bottom, scenes of houses, landslide, ruins, clogged, ponding and trees are shown.

**Table 9.** Test results of the optimal PEMSR model. This model is trained after data augmentation by a factor of 8 and data balancing.

| Metrics | Houses | Landslide | Ruins | Ponding | Trees | Clogged |
|---------|--------|-----------|-------|---------|-------|---------|
| Precision | 84.01% | 91.55% | 90.42% | 84.00% | 83.33% | 72.91% |
| Recall | 92.12% | 93.10% | 87.66% | 83.33% | 72.22% | 71.43% |
| F1 | 87.88% | 92.32% | 89.02% | 83.66% | 77.34% | 72.16% |

## 4. Discussion

The proposed PEMSR model realizes six types of postearthquake scene recognition. Then, the model is optimized, and the recognition performance is improved. Through several experiments, the PEMSR model demonstrates two advantages. First, the PEMSR model based on SSD with transfer learning outperforms the HOG+SVM method in recognition. Second, data augmentation and balancing overcome the problems caused by insufficient and imbalanced datasets, which improves the accuracy of the PEMSR model on postearthquake scene recognition. In addition, the model facilitates the identification of areas that merit further study.

### 4.1. PEMSR Model with Transfer Learning Outperforms Other Methods

According to Table 6, the PEMSR model shows a higher recognition efficiency compared with the traditional HOG+SVM machine learning method: the average detection time required is only 0.4565s, while that of the HOG+SVM method is 8.3472s, and its overall recognition accuracy is higher. The application of the transfer learning method results in a significant improvement on the task of training a model with insufficient samples. The PEMSR model that is proposed in this paper is based on the SSD method which uses the transfer learning strategy to reduce the required training sample data volume. As Table 10 shows, the overall accuracy for each type of scene is improved due to the transfer learning strategy, although the average detection time is slightly longer compared to the SSD method. The PEMSR model shows better overall accuracy via transfer learning.

**Table 10.** F1 score results for PEMSR and the SSD method on the original data. ADT refers to the average detection time and the bold values are the best results among the two methods for each scene.

| Method | Houses | Landslide | Ruins | Ponding | Trees | Clogged | ADT |
|--------|--------|-----------|-------|---------|-------|---------|-----|
| SSD | 75.22% | 83.57% | 82.95% | 70.31% | 37.88% | 46.21% | **0.4123 s** |
| PEMSR | **79.17%** | **86.67%** | **87.78%** | **72.73%** | **39.98%** | **50.00%** | 0.4565 s |

### 4.2. Data Augmentation and Balancing Improves the Accuracy of PEMSR Model

Most deep learning methods require sufficient sample data; otherwise, the training performance will be poor or overfitting will occur. In the PEMSR model, data augmentation is used to overcome the problem of poor original samples. Table 11 presents the F1 score results of each postearthquake scene for every augmentation experiment. The overall recognition accuracy of the PEMSR model increases with each data augmentation. However, there are also scenes with very small improvements, such as ruins. On this scene, the recognition performance is close to the optimal recognition performance of the PEMSR model. The ruins recognition result F1 score tends to be stable, with only a few fluctuations in each different experiment, and it is difficult to improve the performance via data augmentation. The imbalanced dataset may bias the model towards the majority class of the sample, whereas the recognition of the minority class of the sample is not satisfactory. Whilst simply increasing the amount of training data may not continue to improve the performance of the model, we consider applying the oversampling method in data augmentation to balance the dataset. The results demonstrate that when our dataset was balanced, the recognition performance of the PEMSR model improved substantially, especially on classes of scenes that occupy small proportions of the original dataset, such as ponding,

trees and clogged. It is concluded that our PEMSR model offers advantages when faced with sample shortages and imbalance.

**Table 11.** F1 score results after each data augmentation of the PEMSR model. SB refers to the F1 score results after data balancing. The bold values are the best results for each scene.

| Times | Houses | Landslide | Ruins | Ponding | Trees | Clogged |
|---|---|---|---|---|---|---|
| Original | 79.17% | 86.67% | 87.78% | 72.73% | 39.98% | 50.00% |
| Twice | 80.00% | 86.24% | 87.34% | 72.79% | 53.33% | 63.16% |
| Four | 80.61% | 85.48% | 88.06% | 74.42% | 57.14% | 60.00% |
| Eight | **88.28%** | 90.37% | 88.02% | 76.08% | 64.10% | 62.06% |
| SB | 87.88% | **92.32%** | **89.02%** | **83.66%** | **77.34%** | **72.16%** |

*4.3. Future Enhancements*

As reported in this paper, the PEMSR model realizes excellent recognition performance in postearthquake scene recognition. However, there is scope for further development and application. For example, this paper does not consider the influence of the image resolution on the recognition performance. In the future, contrast experiments will be conducted through image blurring, and the influence of the resolution on PEMSR model recognition performance will be explored. Besides, the samples of this paper come from a mountainous seismic area, and the recognition effect of the model in the urban building dense area needs further verification. Furthermore, we consider taking the common scenes such as houses, trees and ponding as a type of background sample and analyze their impact on model recognition of scenes caused by earthquakes. At the same time, more types of scenes caused by earthquakes were added, such as ground cracks. Finally, the application of additional knowledge from disaster science to realize hierarchical recognition of postearthquake scenes is a subject for further investigation. For example, ruins can be divided into severe and mild damage classes, which may be more valuable for postearthquake relief and reconstruction.

## 5. Conclusions

This paper proposes a PEMSR model based on the classical SSD detection method that was focused on overcoming the over-reliance on expert visual interpretation and the poor recognition performance of traditional machine learning in postearthquake scene recognition. In the model, a transfer learning and data augmentation strategy are utilized to overcome the dataset insufficiency; moreover, a random oversampling method is used to overcome dataset imbalance. The results of several comparison and optimization experiments demonstrate that the model realizes satisfactory performance in postearthquake scene recognition and that transfer learning, data augmentation and data balancing strategies improve the recognition performance of the PEMSR model. Although there are many opportunities for further exploration and development, the results thus far are encouraging in the field of postearthquake scene recognition.

**Author Contributions:** Zhiqiang Xu and Yumin Chen conceived and designed the experiments; Fan Yang and Tianyou Chu processed the data; Zhiqiang Xu, Fan Yang and Hongyan Zhou performed the experiments; Zhiqiang Xu and Yumin Chen wrote the paper; Tianyou Chu and Hongyan Zhou prepared the figures for the paper. All authors have read and agreed to the published version of the manuscript.

**Funding:** This research received no external funding.

**Acknowledgments:** This work is financially supported by National Key R&D Program of China: (grant number 2018YFB0505302); and the National Nature Science Foundation of China: (grant number 41671380).

**Conflicts of Interest:** The authors declare no conflict of interest.

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
