# Peer review of "A Postearthquake Multiple Scene Recognition Model Based on Classical SSD Method and Transfer Learning"

_ijgi, doi:10.3390/ijgi9040238_

Round 1
Reviewer 1 Report
This paper is interesting, readable and has a good structure. Consequently, the article draft should be major revision for publishing in IJGI. Some details can be listed as below:
Abstract:
- This part isn’t complete. I don’t find main results abstract. Consequently, you should describe that.
Introduction:
- The main subject of this manuscript is analyzing damage of post-earthquake. So, I think that write about statistics on earthquake damage. Also, manuscript don’t have study area part! it is more important expiation feature of study area and historical earthquake.
- There are several satellites with different bands and resolutions for analyzing damage of post-earthquake. But, you write about RGB and google map. There are several papers and research about using different such as sentinel1, 2 and etc. please add more references about that.
Materials and Methods:
- PEMSR model’s workflow isn’t clear and complete. Please redesign again add detail.
- Which software do you use for segmentation and Classification phase?
- Why do you divide into a training dataset and a test dataset in a 4:1 ratio?
- Finding structure of network is very important. Figure 4 shown structure of PEMSR network. How find this structure?!
- Deep learning need three parts: train, valid, test data. You delete valid data. How do you detect and solve overfitting problem?
- You utilize HOG+SVM in manuscript for comparing with PEMSR. But, HOG+SVM model isn’t clear. please add more references and tune of parameters of HOG+SVM.
Experiments and Results
One of important point in machine learning is speed of process. I don’t find about speed of process or convergence graph of each model!
Discussion and Conclusions:
Those parts are well-write.
I hope this will help you improve your manuscript.
Author Response
Dear Reviewer, Thanks for your comments. We have carefully examined the comments and revised the manuscript accordingly. We would like to express our appreciation to your constructive comments and suggestions. Detailed corrections are listed point by point. Point 1: Abstract isn’t complete. I don’t find main results abstract. Consequently, you should describe that. Response 1: We are sorry this part is incomplete. The abstract was supplemented as you suggested. (page.1 line 20-24). Point 2: The main subject of this manuscript is analyzing damage of post-earthquake. So, I think that write about statistics on earthquake damage. Response 2: Thanks for you advise. In the introduction, We took the Ludian earthquake as an example, and added some statistics to showed the damage caused by the earthquake. (page.1 line 29-34). Point 3: Also, manuscript don’t have study area part! it is more important expiation feature of study area and historical earthquake. Response 3: Thanks for you advise. In the Experiments and Results part, We added a section of study area to give more descriptions. (page .9 line 362-371). Point 4: There are several satellites with different bands and resolutions for analyzing damage of post-earthquake. But, you write about RGB and google map. There are several papers and research about using different such as sentinel1, 2 and etc. Please add more references about that. Response 4: Thanks for you advise. We added more references about that. (page .2 line 46-60). Point 5: PEMSR model’s workflow isn’t clear and complete. Please redesign again add detail. Response 5: We are sorry the figure is unclear and incomplete. We redesigned Figure 1 to make it clear and intuitive. (page. 3 line 116-119). Point 6: Which software do you use for segmentation and Classification phase? Response 6: We segmented the images by ArcGIS Desktop, and classified by ourselves. In the manuscript, we added relevant descriptions. (page .4 line 132-137). Point 7: Why do you divide into a training dataset and a test dataset in a 4:1 ratio? Response 7: Considering the limited amount of data in the original data set, we want to ensure sufficient training set to avoid overfitting, at the same time, we also want to ensure the confidence of the test results, so we think 4: 1 is a suitable ratio. We added relevant descriptions at (page .4 line140-142). Point 8: Finding structure of network is very important. Figure 4 shown structure of PEMSR network. How find this structure?! Response 8: Our PEMSR model is based on the SSD model, and the SSD model is transformed on the classic VGG16 network structure. SSD extracts multi-scale features by adding multi-scale convolutional layers, our PEMSR model inherits the advantages of this structure and fine-tunes network parameters by learning from our earthquake dataset. (page. 7 line 266-282) Point 9: Deep learning need three parts: train, valid, test data. You delete valid data. How do you detect and solve overfitting problem? Response 9: I'm sorry I didn't make it clear in the manuscript. Actually, we didn’t delete valid data. When we transformed the samples to VOC dataset, we only need to divide the dataset into training or test dataset, and determined their ratio is 4: 1, we don’t need to allocate a validation dataset in advance. In each training process, we randomly selected 1/8 training dataset as the validation dataset through preset parameters. It means that our training: validation: test dataset has a ratio of 7: 1: 2. To avoid similar confusion, we modified relevant descriptions in manuscript. (page. 4 line 140, page. 7 line 259 line 263). Point 10: You utilize HOG+SVM in manuscript for comparing with PEMSR. But, HOG+SVM model isn’t clear. please add more references and tune of parameters of HOG+SVM. Response 10: I'm sorry this part is not clear. Previously, we considered that HOG+SVM was not the method mainly explained in our paper, so it did not been introduced too much. According to your advice, we added more explanations and references about it. (page. 9 line 336-344). Point 11: One of important point in machine learning is speed of process. I don’t find about speed of process or convergence graph of each model! Response 11: According to your advice, we retrained the optimal PEMSR model and recorded the loss value convergence in Figure 6. (page. 12 line 432-445)
Reviewer 2 Report
The manuscript was well organized. The authors proposed a transfer learning model to classify the post-event optical images. The results were verified by comparing to two other methods and showed good accuracy. The influence of data volumes was well discussed.
The only doubt is the versatility of the proposed model. Since the input images were one earthquake event in the mountain area. The houses were sparse and easy to be identified. I wonder how the model works in the urban area.
Some minor comments:
End of the second paragraph in P. 2: "Although the SSD method is slower than YOLO, its accuracy exceeds that of YOLO by 10% and is comparable to that of the RCNN series." Please add the reference for this sentence.
Table 2 "Number of instances of each scene" According to the following contents, the number in the table looks like the total number from two scenes. Please confirm it.
2.2.1 first paragraph: "The segmented images are randomly divided into a training dataset and a test dataset in a 4:1 ratio". The data augmentation was carried after the division, which means the test dataset was also augmented. Please indicate the final number of images for the training and testing. The augmentation of the test dataset is meaningless.
The samples for ponding, tress and clogged were limited compared to those of houses and ruins. After the data augmentation, the numbers became 50 times. Overfitting is considered. Is it possible to add one other scene including more samples of those three classes?
Author Response
Dear Reviewer, Thanks for your comments. We have carefully examined the comments and revised the manuscript accordingly. We would like to express our appreciation to your constructive comments and suggestions. Detailed corrections are listed point by point. Point 1: The only doubt is the versatility of the proposed model. Since the input images were one earthquake event in the mountain area. The houses were sparse and easy to be identified. I wonder how the model works in the urban area. Response 1: The reason we chose this case is not only to detect damaged houses, but also because mountainous areas are more likely to cause secondary disasters such as landslides, which is one of the scenes we hope to identify. Thanks for your advice. The question you mentioned that how the recognition effect of the model for urban area will be added to our future research discussions. (page. 15 line 498-500). Point 2: End of the second paragraph in P. 2: "Although the SSD method is slower than YOLO, its accuracy exceeds that of YOLO by 10% and is comparable to that of the RCNN series." Please add the reference for this sentence. Response 2: We are sorry we made a mistake in this statement, we modified and added references about it. Thanks for your comment! (page. 2 line 82-84). Point 3: Table 2 "Number of instances of each scene" According to the following contents, the number in the table looks like the total number from two scenes. Please confirm it. Response 3: Table 2 shows the number of six categories of scenes. And these images came from we segmenting the original large images to 300*300 by ArcGIS Desktop and manually dividing them. (page. 4 line 132-138). Point 4: 2.2.1 first paragraph: "The segmented images are randomly divided into a training dataset and a test dataset in a 4:1 ratio". The data augmentation was carried after the division, which means the test dataset was also augmented. Please indicate the final number of images for the training and testing. The augmentation of the test dataset is meaningless. Response 4: Thanks for your comment. We agree with what you said is very reasonable. Generally, in the deep learning process, we will not do data augmentation on the test dataset. However, considering that our original dataset is too small, the amount of data that can be allocated to the test dataset is smaller. In order to reduce the contingency of the test results, we choose to perform data augmentation on the test dataset at the same time. Besides, we believe that data augmentation to the test dataset may decrease the test accuracy, but it can reflect the detection ability of the PEMSR model in the case of multiple complex augmentation on the test dataset, which is also a reflection of the excellent detection effect of the model. Final number of images for the training and testing were indicated at page. 6 line 238-240. Point 5: The samples for ponding, tress and clogged were limited compared to those of houses and ruins. After the data augmentation, the numbers became 50 times. Overfitting is considered. Is it possible to add one other scene including more samples of those three classes? Response 5: Thanks for your comment, your suggestion is extremely valuable. In our experiment, we considered that the identification of clogged is helpful for post-earthquake relief, so we considered it separately. In the post-earthquake scene recognition, it is obvious that the recognition priority for ruins, landslide, clogged, and other earthquake-induced scenes is higher than common scenes like houses, trees. We considered that it is necessary to identify the scenes caused by earthquakes separately. Maybe we can divide these common scenes into a class of recognition targets according to your suggestions. At the same time, we will also try to add more earthquake-induced scenes such as ground crack. That will be considered in our future work. (page .15 line 500-503)
Round 2
Reviewer 1 Report
I have read the revised version of the manuscript carefully. The authors have dealt with the comments and suggestions of reviewers in a highly satisfactory and constructive manner. The revised manuscript clearly meets the standards that have been required by reviewers. I propose the acceptance of the manuscript.
Author Response
Point1: I have read the revised version of the manuscript carefully. The authors have dealt with the comments and suggestions of reviewers in a highly satisfactory and constructive manner. The revised manuscript clearly meets the standards that have been required by reviewers. I propose the acceptance of the manuscript.
Response1: Thanks to the reviewers for your recognition and dedication. Thank you for your constructive comments on this manuscript. We are very glad that this manuscript was accepted by your journal. Thanks again!
